# Simple Growth–Metabolism Relations Are Revealed by Conserved Patterns of Heat Flow from Cultured Microorganisms

**DOI:** 10.3390/microorganisms10071397

**Published:** 2022-07-11

**Authors:** Karim Fahmy

**Affiliations:** 1Helmholtz-Zentrum Dresden-Rossendorf, Institute of Resource Ecology, Bautzner Landstrasse 400, 01328 Dresden, Germany; k.fahmy@hzdr.de; 2Cluster of Excellence Physics of Life, Technische Universität Dresden, 01062 Dresden, Germany

**Keywords:** bacteria, enzyme kinetics, heavy metals, isothermal microcalorimetry, Michaelis–Menten, toxicity

## Abstract

Quantitative analyses of cell replication address the connection between metabolism and growth. Various growth models approximate time-dependent cell numbers in culture media, but physiological implications of the parametrizations are vague. In contrast, isothermal microcalorimetry (IMC) measures with unprecedented sensitivity the heat (enthalpy) release via chemical turnover in metabolizing cells. Hence, the metabolic activity can be studied independently of modeling the time-dependence of cell numbers. Unexpectedly, IMC traces of various origins exhibit conserved patterns when expressed in the enthalpy domain rather than the time domain, as exemplified by cultures of *Lactococcus lactis* (prokaryote), *Trypanosoma congolese* (protozoan) and non-growing *Brassica napus* (plant) cells. The data comply extraordinarily well with a dynamic Langmuir adsorption reaction model of nutrient uptake and catalytic turnover generalized here to the non-constancy of catalytic capacity. Formal relations to Michaelis–Menten kinetics and common analytical growth models are briefly discussed. The proposed formalism reproduces the “life span” of cultured microorganisms from exponential growth to metabolic decline by a succession of distinct metabolic phases following remarkably simple nutrient–metabolism relations. The analysis enables the development of advanced enzyme network models of unbalanced growth and has fundamental consequences for the derivation of toxicity measures and the transferability of metabolic activity data between laboratories.

## 1. Introduction

Monitoring cellular replication allows researchers to better understand the link between microbe metabolism and growth. In its simplest form, this relation has found its analytical expression in the Monod equation, which describes the nutrient dependence of the initial exponential growth rate of a bacterial culture (balanced growth), marking the beginning of quantitative growth analyses in microbiology [1]. Since then, the development of instrumentation for the real-time monitoring of cell numbers has not ceased [2,3,4], nor has the analysis and formulation of mathematical growth models [5,6], the proteome-based modeling of balanced growth [7,8], as reviewed in refs. [9,10,11], or the physical interpretation and modification of the Monod equation [12,13]. In contrast to the indirect assessment of metabolism by cell number measurements, isothermal microcalorimetry (IMC) records metabolic activity directly as the thermal metabolic power generated by microorganisms as a consequence of their overall exothermic biochemical reactions [14,15,16,17,18]. Therefore, IMC raw data scale directly with the absolute growth rate *r = dN/dt* at a signal to noise ratio currently unachievable by turbidity assays [19], from which R needs to be determined from the first derivative of the raw data and, thus, to severe noise enhancement. IMC is increasingly used in environmental [20,21,22,23], pharmacological [24,25] and medical [26,27,28,29] research and mostly addresses the potency of inhibitors of bacterial growth.

Importantly, IMC provides unprecedented opportunities for the study of complex growth–metabolism relations, as exemplified by the analysis of the physiological causes of Kleiber’s law in *Plathelminthes* [30] or the metabolic arrest of *Nematodes* [31]. As already seen with microorganisms, an unexpected richness in the detail of the time-dependent heat flow curves is commonly encountered but poorly understood. Such sub-structures typically appear close to the maximal absolute growth of a culture, indicative of metabolic adaptation during unbalanced growth [9,10,11] and are not resolved by current assays reporting cell numbers. In the context of quantitative growth–metabolism models, it is thus appealing to use IMC’s high sensitivity (10 nW) and high sampling rate (usually one data point per second) to investigate metabolic heat flow patterns of cultured microorganisms. For example, the original restriction of the Monod equation to the exponential culture growth has little physiological foundation. Rather, it has been motivated by its simplicity in describing this particular phase of constant culture conditions, i.e., “balanced growth”. There is no biological reason why the nutrient dependence of metabolism should abruptly change after exponential growth. This suspicion has motivated the (re-)evaluation of the metabolic thermal power curves over the full “life span” of cultured cells focusing on the applicability of the Monod-like nutrient dependence of growth beyond the exponential phase. For this purpose, a “dynamic adsorption reaction-thermogram simulation” (dAR-TS) is developed here to decompose metabolic heat curves from cultured cells into successive “canonical thermograms”. The introduced formalism is not restricted to IMC data analyses but is described here for heat flow measurements because they provide the required data quality and time resolution to express metabolism in dependence of the total biomass. The approach was further stimulated by the similarity of heat flow patterns from various microorganisms in line with conserved fundamental physiological traits [32,33,34]. In fact, the analyses of previously published and novel IMC data reveal that “canonical thermograms” describe metabolic heat production of cultured cells with high accuracy, enabling novel experimental approaches to evaluate complex biological growth models [35]. The quantitative description of growth inhibition is further elaborated here to improve transferability of toxicity data between labs. The analysis uses biological parameters which describe a hyperbolic nutrient dependence of growth with correlates in the original descriptive Monod approach, but also in line with more advanced mechanistic models, which result in hyperbolic dependencies as well. However, in contrast to the latter models, growth described by dAR-TS holds for unbalanced conditions as well. The prevalence of the underlying simple nutrient dependence of metabolic activity beyond exponential growth is revealed when the increase in catalytic capacity with culture growth is explicitly considered. This has been done here by exploring the relationship between metabolic thermal power and its integral, the total heat release from replicating prokaryotic (*Lactococcus lactis*) and eukaryotic (*Trypanosoma congolese*), as well as non-dividing cells (*Brassica napus*, plant callus culture). The hyperbolic dependence of metabolic activity on nutrient concentration, formally introduced by Monod to describe the balanced growth of bacterial cultures appears to be the rule, rather than an exception restricted to balanced growth only.

## 2. Materials and Methods

### 2.1. Heat Flow Curves from L. lactis at Different Temperatures

The strain *Lactococcus lactis* subsp. *cremoris* NZ9000/pNZ8148 (Nizo Food Research BV, Ede, The Netherlands) was grown at pH 7.2 in M17 medium containing (per liter) 5 g Neopeptone, 5 g Bacto Soyton, 5 g beef extract, 2.5 g yeast extract, 0.5 g ascorbic acid and 1 g Na_2_-β-glycerophosphate · 5 H_2_O. After autoclavation, 1 mL of 1 M MgSO_4_ and 20 mL of 50% glucose were added from sterile-filtered solutions. IMC was performed with a TAMIII instrument (Waters GmbH, Eschborn, Germany). Calorimetric ampoules were filled with 4 mL medium and inoculated with 20 µL of a main bacterial culture, which was pre-diluted with medium to OD_600_ = 0.1. Thermal power was monitored continuously over 50 h at 20 °C or 25 °C. Cultivation and IMC data acquisition for all other experiments have been published for *L. lactis* [36], *B. napus* [37] and *T. congolese* [38].

### 2.2. Thermogram Simulations

Experimental IMC traces were fitted to “canonical thermograms” according to the sequence of data treatments explained below. The fits are based exclusively on the experimentally determined values of the peak metabolic activity and the released heat at which this maximum is reached. If sufficient substructure is resolved in an IMC trace, multiple of such data pairs can be obtained and, if improperly resolved, optimized by dAR-TS to minimize the standard deviation of the “canonical thermogram” from the measured IMC trace in a selected data range. The following abbreviations are used:

$E∶=\frac{\Delta N}{\Delta H}$, growth efficiency (increase of metabolizing units per Joule of released heat)

*H*: released metabolic heat

*H*_0_: heat equivalent of total amount of nutrients turned over in a specific metabolic state

*H_f_*: heat equivalent of the amount of nutrients supporting half-maximal growth

*H_m_*: released metabolic heat by the time *P_m_* is reached

$H_{\emptyset }\left(t\right)∶=\frac{1}{t}\overset{t}{\underset{0}{\int }}Hdt$, time-averaged metabolic heat released

*N*_0_: initial number of metabolizing units

*P*: thermal power

*P_m_*: maximal thermal power of a thermogram

$p_{0}$: substrate-saturated thermal power per metabolizing unit

Θ: metabolic load (0≤Θ≤1)

$r_{0}∶=p_{0}\cdot E$, substrate-saturated maximal growth rate

$r_{i}∶=$ initial growth rate

### 2.3. Relationship between Experimental Observables and Thermogram Parameters for Simulations

The fundamental equation of IMC is:(1a)$$P\left(H\right)=p_{0}\cdot \left(N_{0}+E\cdot H\right)\cdot \Theta \left(H\right)$$
with
(1b)$$\theta \left(H\right)=\frac{H_{0}-H}{H_{0}-H+H_{f}}$$
and the initial growth rate
(1c)$$r_{i}=\frac{dP}{dH}\left(0\right)=r_{0}\cdot \frac{H_{0}}{\sum^{\;}}=r_{0}\cdot \theta \left(0\right).$$

From the maximum condition $\frac{dP}{dH}=0$ and with the definition $\sum^{\;}∶=H_{0}+H_{f}$, the thermogram peak *P_m_* and the total released heat *H_m_* at the peak are given by
(2a)$$P_{m}=r_{0}\left[\sum^{\;}+H_{f}+\frac{N_{0}}{E}-2\cdot \sqrt{H_{f}\cdot \left(\sum^{\;}+\frac{N_{0}}{E}\right)}\;\right]$$
(2b)$$H_{m}=\sum^{\;}-\sqrt{H_{f}\cdot \left(\sum^{\;}+\frac{N_{0}}{E}\right)}$$

From Equations (2a) and (2b), follow the three relations:(3a)$$\frac{P_{m}}{r_{0}}=2H_{m}-H_{0}$$
(3b)$$\frac{P_{m}}{r_{0}}=\frac{H_{m}^{2}}{\sum^{\;}}$$

For $\frac{N_{0}}{E}\ll \;\sum^{\;}$, i.e., for small inoculates assumed in all of the following equations, the thermogram “shape parameter” *s_p_* is defined as:(3c)$$s_{p}∶=\frac{P_{m}}{r_{i}\cdot H_{m}}=\frac{H_{m}}{H_{0}}=1+\frac{H_{f}}{H_{0}}-\frac{H_{f}}{H_{0}}\cdot \sqrt{1+\frac{H_{0}}{H_{f}}}$$
from which follows the property $1>s_{p}>0.5$ for all positive values of *H_f_*.

### 2.4. Determination of “Canonical Thermogram” Parameters from IMC Traces

The Equations (3a)–(3c) enable the derivation of “canonical thermograms” from IMC traces. Simulation starts with the selection of a data pair *P_m_* and *H_m_* based on well-resolved thermogram peaks or “shoulders”. With the definition $\alpha ∶=\frac{P_{m}}{H_{m}}$ and the use of the thermogram “shape parameter” in Equations (3a) and (3b), the “canonical thermogram” parameters are obtained as:(4a)$$H_{0}=\frac{H_{m}}{s_{p}}$$
(4b)$$r_{0}=\frac{s_{p}}{2\cdot s_{p}-1}\cdot \alpha$$
(4c)$$H_{f}=\frac{{\left(s_{p}-1\right)}^{2}}{s_{p}\cdot \left(2\cdot s_{p}-1\right)}\cdot H_{m}$$
(4d)$$\theta \left(0\right)=\frac{2\cdot s_{p}-1}{s_{p}^{2}}$$

Here, Equation (4b) was used in Equation (3b) to derive Equation (4c). Equation (4d) follows from Equations (4a) and (4c). The “canonical thermogram” is obtained by substituting the above parameters in Equation (1a) (with *N*_0_ = 0). During curve fitting, the thermogram “shape parameter” is varied between 0.5 and 1 to minimize the standard deviation between the “canonical thermogram” and the original IMC trace. The procedure is repeated for neighboring *P_m_*, *H_m_* data pairs until the best fit to a selected data range is achieved.

### 2.5. Constructing a “Rate Plot” from “Canonical Thermogram” Parameters

The “rate plot” displays the initial exponential growth rate *r_i_* as a function of the initial “metabolic load” Θ(0). The latter is expressed by the thermogram “shape parameter” (Equation (4d)) and equals the fraction of the initial growth rate *r_i_* relative to the maximal (nutrient-saturated) growth rate *r*_0_ in any metabolic phase (Equation (1c)) analogous to a Langmuir adsorption isotherm.

The “rate plot” visualizes growth-inhibiting and -promoting effects as displacements of data pairs [Θ(0); *r_i_*] relative to a reference point defined by an initial “metabolic load” of 0.5 and an initial growth rate of 0.5 (corresponding to *r*_0_ = 1, Equation (1c)). The following steps produce the data pair $\left[{\;\bar{\Theta }}^{+}\left(0\right);{\bar{r}}_{i}^{+}\right]$. It is obtained from dAR-TS for the thermogram recorded with a growth-affecting additive (labeled with “+”) and extrapolated to an initial “metabolic load” of 0.5 (labeled with “¯”).

### 2.6. Normalization of Metabolic Load

The normalization of the initial “metabolic load” of the reference thermogram is equivalent to choosing standard conditions (nutrient concentration and culture volume) such that the complete consumption of the initial nutrient in the reference measurement would release the total heat *H_f_* (Equation (1b)). Therefore, the normalized condition for the thermogram with additive requires ${\bar{H}}_{o}^{+}=H_{f}$ and the normalized initial metabolic load is
(5)$${\bar{\theta }}^{+}\left(0\right)=\frac{H_{f}}{H_{f}+H_{f}^{+}}.$$

The values of *H_f_* and $H_{f}^{+}$ are taken from fits to the experimental data according to Equation (4d).

### 2.7. Normalization of the Initial Growth Rate

Normalization of the initial growth rate of the culture with additive relative to the reference culture makes use of the maximal growth rates obtained from both cultures by dAR-TS (Equation (1c)):(6)$${\bar{r}}_{i}^{+}=\frac{r_{0}^{+}}{r_{0}}\cdot {\bar{\theta }}^{+}\left(0\right)=\frac{r_{i}^{+}}{\theta^{+}\left(0\right)}\frac{\theta \left(0\right)}{r_{i}}{\bar{\theta }}^{+}\left(0\right)$$

The “metabolic load” under standard conditions for the measurement with additive is obtained by substituting in Equation (5) the sample and reference parameters defined in Equation (4c):(7)$${\bar{\theta }}^{+}\left(0\right)=\frac{1}{1+\frac{H_{m}^{+}\cdot {\left(1-s_{p}^{+}\right)}^{2}\cdot \left(2\cdot s_{p}-1\right)\cdot s_{p}}{H_{m}\cdot {\left(1-s_{p}\right)}^{2}\cdot \left(2\cdot s_{p}^{+}-1\right)\cdot s_{p}^{+}}\;}$$

### 2.8. The Closed Form of Heat Evolution in dAR-TS

With the definitions *G = H*_0_
*− H*, *r*_0_
*= p*_0_*∙E* and *p_N_*_0_
*= p*_0_*∙N*_0_, Equation (1a) can be written as
(8)$$-G^{\prime }=(p_{N0}+r_{0}H)\frac{G}{G+H_{f}}$$
from which follows
(9)$$\ln G=-\frac{p_{N0}}{H_{f}}t-\frac{r_{0}}{H_{f}}\overset{t}{\underset{0}{\int }}Hdt-\frac{G}{H_{f}}+c$$

The integration constant $c=H_{0}\cdot e^{\frac{H_{0}}{H_{f}}}$ ascertains the condition *G*(0) *= H*_0_ and one obtains
(10)$$G=H_{0}\cdot e^{\frac{H_{0}}{H_{f}}}\cdot e^{\frac{-p_{N0}\cdot \;t-r_{0}\int_{0}^{t}Hdt-G}{H_{f}}\;}$$
from which the closed form follows as:(11)$$\frac{H_{0}-H}{H_{f}}=W\left(\frac{H_{0}}{H_{f}}\cdot e^{\frac{H_{0}-r{\;}_{0}\cdot H_{\emptyset }\left(t\right)\cdot t}{H_{f}}}\right)$$

Here, the definition $H_{\emptyset }\left(t\right)∶=\frac{1}{t}\overset{t}{\underset{0}{\int }}Hdt$ has been used for the time-averaged total heat release.

## 3. Results

The similar shape of IMC traces recorded from different cultured microorganisms motivated the attempt to describe heat (enthalpy) release patterns analytically. Revisiting IMC data from this and other labs reveals that maximal growth rate *r_i_*, maximal thermal power *P_m_* and the amount of released heat *H_m_* at maximal growth of cultured cells determine a thermogram “shape parameter” *s_p_* from which growth–metabolism relations can be derived (Methods). The analysis developed in this text relies on the trivial recognition of the inherent nutrient-dependence, rather than the time-dependence of metabolism. When expressed in the enthalpy domain, nutrient–metabolism relations can be formulated, solved analytically and transformed back into the time domain. This is shown here for a Langmuir adsorption reaction model. It describes metabolism as nutrient adsorption, followed by exothermic chemical turnover and growth according to the increase in adsorption capacity. The interdependence of both reproduces thermograms from diverse cultured cells. Section 1 and Section 2 introduce the enthalpy-based parametrization of IMC data. Section 3 exemplifies comprehensive analyses of thermograms from bacteria, protozoans and plant cells. Section 4 uses dAR-TS to standardize IMC evaluations on growth inhibition and stimulation. The reproduction of calorimetric results within the signal to the noise ratio of the raw data lends strong support to a general trait of growth–metabolism relations of cultured cells covering balanced conditions as well as maximal growth and metabolic decline.

### 3.1. Transformation of Metabolic Thermal Power from Time- to Enthalpy-Domain

The concept developed here for the analysis of metabolic heat flow data is rooted in evaluating the heat-dependence, rather than the time-dependence of metabolic activity. As total heat release (enthalpy) scales with biomass, growth–metabolism relations are more directly reflected in this representation than in the time domain. However, thermal metabolic power *P* is typically displayed as a function of time *t* as shown in Figure 1a for a culture of *Lactococcus lactis* grown in the presence of different concentrations of copper [36]. The metal delays growth and reduces the maximal power *P_m_* in a concentration-dependent manner. The heat flow generated by cultured microorganisms can be expressed by the product of the cell number (*N*) and the thermal power per cell (*p*). The latter is expressed here as the product of *p*_0_, the maximally achieved thermal power per cell and Θ, the “metabolic load”, which represents the relative activity of metabolism (between zero and unity), dependent on the culture conditions (Equation (12)). For a given composition of nutrients turned over in a specific biochemical pathway, *p*_0_ is constant, whereas all other quantities change over time:(12)$$P\left(t\right)=p_{0}\cdot \;N\left(t\right)\cdot \Theta \left(t\right)$$
(13)$$H\left(t\right)=\overset{t}{\underset{0}{\int }}P\left(t\right)dt$$

By definition, the released heat *H* (enthalpy) is the time integral over *P* (Equation (13)). Figure 1b displays *P* as a function of *H*, the designated “enthalpy plot”. Exponential growth corresponds to a linear dependence on *H* (mathematically equivalent to a logarithmic plot in the time domain). The transformation reveals more similarity between the original thermograms because the transition from an initial to a second metabolic state, visible from the thermogram peak and shoulder, occurred at different times but at almost the same amount of released heat (ca.3 J). The apparent physiological relevance of *H* rather than *t* is plausible because *H* scales with the amount of consumed nutrients (and the corresponding amount of biomass).

Figure 1c shows thermograms from the same bacterial strain in the presence of the toxic salt uranyl nitrate. The initial, almost metal-insensitive, exponential growth was followed by a uranyl-sensitive metabolic state. This is seen clearer in the “enthalpy plot” in Figure 1d, where the initial linear rise is barely affected, whereas the peak activity in the second metabolic state is attenuated and shifts to smaller enthalpies with increasing uranyl concentration. These different mechanisms of copper and uranyl toxicity [36] exemplify a toxic effect predominantly on *p*_0_ for copper or on “metabolic load” Θ(*H*) for uranyl. The qualitative complementarity of metal toxicities in the same bacterial strain demonstrates that at least two measures are required to describe toxicity. In particular, the data reveal the weakness of deriving toxicity measures exclusively from exponential growth rates.

### 3.2. Simulation of “Canonical Thermograms”

Metabolism in its simplest representation removes chemical compounds from solution, followed by enzymatic turnover. Formally, the process resembles the adsorption of solutes to a catalytic surface. According to the scheme in Figure 2, the metabolic load Θ would express the surface coverage of such an apparent reactive surface, described here by a Langmuir isotherm as shown in Figure 2 (Equation (1b)), where the concentration of “solute”, in fact a complex mixture of nutrients, scales with the expression *H*_0_
*− H*(*t*). The heat equivalent of the initial amount of nutrients is given by *H*_0_, whereas *H_f_* determines the condition for the half-maximal metabolic activity, similar to a Langmuir adsorption coefficient. In this model, cell division entails an incremental growth of the total adsorptive surface *A* in proportion to the amount of heat released between divisions. In contrast to the classical Langmuir adsorption with a static surface area, the proposed model features a growing adsorptive surface, enabling a “dynamic adsorption reaction thermogram simulation” (dAR-TS). Substituting the term *r*_0_∙*H* for *p*_0_∙*N* leads to the analytical expression of the enthalpy plot (Figure 3c, Equation (1a)).

Equation (1a) is used to analyze IMC enthalpy plots of cultures grown from small inoculates which generate “zero” thermal power initially (see Methods for the general IMC equation). Figure 3a–d show the generation of “canonical thermograms” according to Equation (1a) and compare the results for two different maximal growth rates *r*_0_ with the same Langmuir isotherm (“metabolic load” Θ). The “enthalpy plots” differ by a constant factor (Figure 3c) and can be transformed back to the time domain by numeric integration (Figure 3d). The obtained “canonical thermograms” show qualitatively that copper toxicity complies with a reduced maximal growth rate *r*_0_, leading to prolonged heat release at reduced power. Figure 3e–h show the complementary situation, i.e., a reduction of the “metabolic load” by an increase of *H_f_*. The obtained “enthalpy plots” show again a reduced peak metabolic activity but, importantly, also a shift of the peak to lower heat values similar to the effect of uranyl on the second metabolic phase of the *L. lactis* culture. Thus, dAR-TS reproduces key features of different metal toxicities and relates these to physiologically meaningful quantities; for example, the maximal growth rate and an apparent nutrient affinity expressed by the ”metabolic load” Θ.

### 3.3. Modeling Thermograms from Cultured Cells of Different Origin

In order to validate dAR-TS, thermograms of different origin have been analyzed to identify the largest possible data ranges for which the dependence of the per cell metabolic thermal power on nutrient concentration exhibited the hyperbolic shape predicted by Equation (1b). The results for different cell types and culture conditions are presented. Metal toxicity is considered more specifically because it reveals the physiologic relevance of distinguishing between growth rate and “metabolic load”, which are distinctly affected by different metal ions.

#### 3.3.1. Bacterial Growth Curves

Figure 4 shows the result for uranyl toxicity in *L. lactis* by the independent approximation of the first and second metabolic phases with a pair of parameters r_0_ and H_f_ for either phase. The dotted lines exhibit the “canonical thermogram” shape according to Figure 3c,g. The color-coded areas mark the fitted data ranges, where the contour lines overlap with the raw data within the signal to noise ratio of the IMC measurement. Outside these ranges, the curves are shown only for completeness. Importantly, the IMC data are not modeled by superimposed traces but by consecutive “canonical thermograms” with altered nutrient-dependency of growth, thereby accounting for metabolic adaptation to altered medium conditions (switch to a different carbon source, decrease in the pH during fermentation etc.). In the given example, the two descriptive dAR-TS parameters adopt the values displayed above the thermograms. Uranyl exerted its toxic effect in the second metabolic state predominantly on *H_f_*, which increased about ten-fold (leading to lower metabolic load). The corresponding decrease in apparent nutrient affinity was partly compensated by a ca. three-fold increase of the maximal growth rate *r*_0_, most likely via the up-regulation of enzymes specific to this phase.

The two descriptors of bacterial metabolism, *r*_0_ and Θ, are appropriate to reproduce the inherent complexity of growth-inhibition, suggesting that dAR-TS may also be a predictive tool. The high data quality allows for the validation of dAR-TS with IMC traces for which the outcome of the two parameters may be anticipated. For this purpose, a gentle temperature change was chosen to affect bacterial growth rates (*r*_0_) but not metabolic pathways (Θ). The enthalpy plots of thermograms recorded with *L. lactis* at 20 °C and 25 °C are compared in Figure 4c,d. Although similar in shape, the plots are not related by simple scaling factors. The initial growth phase appears less enhanced by the temperature rise and the released heat at which the growth phase transitions occurred differ non-intuitively between the two experiments (arrows). More remarkable is the result of dAR-TS, which verifies the expected increase of maximal growth rates for all three phases despite the complex thermogram shapes. Importantly, the three phases exhibited negligible temperature-induced variation of their parameter *H_f_*. Thus, the results lend strong support to the physiological significance of the two descriptors in accordance with a thermally increased metabolic rate of an otherwise unaffected microbial physiology.

#### 3.3.2. Protozoan Growth Curves

dAR-TS was further performed with the protozoan *Trypanosoma congolese* using previously published thermograms [38]. In this case, growth was analyzed for different initial inoculates under otherwise identical conditions. The data provide a test for the analysis of thermograms under conditions that would affect neither *r*_0_ nor *H_f_*, but yet resulted in differently shaped thermograms. Figure 5 shows the dissection of the heat flow curves into two “canonical thermograms” for which *H_f_* and the maximal growth rate *r*_0_ were derived from measurements with inoculates of either 2000 or 20 cells in a 2 mL culture medium. For the two metabolic phases, only the altered initial cell number was entered into Equation (1a) and the value *H*_0_ was adjusted as the only varied parameter. The differences between the thermograms are almost exclusively the consequence of the slightly different amounts of total nutrients (different values for *H*_0_) transferred with the increasing inoculate numbers.

The metabolism of *T. congolese* is instructive for another reason. It exhibited only two growth phases without the “asymptotic tailing” of metabolic heat flow seen in the “enthalpy plot” of many bacterial cultures at a late stage of growth (Figure 1 and Figure 4). The rather abrupt decline of metabolic thermal power in the “enthalpy plot” with these protozoans is actually predicted by the Langmuir adsorption reaction model (Figure 3c,g). It is represented in an almost ideal form by the *T. congolese* culture and indicates that the metabolic pathway was strictly conserved during the entire growth, lacking adaptive processes in response to nutrient depletion seen with many prokaryotic cultures. Except for a 12 h gap during which metabolism reoriented, 300 h of culture growth have been simulated. Predicting the decline of metabolism correctly with the same two parameters that also describe the balanced initial growth is a salient strength of dAR-TS.

#### 3.3.3. Metabolism in Cultured Plant Cells

For the purpose of testing more general applicability, dAR-TS was also performed with thermograms from plant cell cultures, where metabolism declined monotonously without growth [37]. Equation (1a) was used with the condition of zero change in cell number per Joule of released heat (*E =* 0). In this case, the “enthalpy plot” is predicted to follow the Langmuir isotherm defined by Equation (1b) (Figure 3a), which is confirmed by dAR-TS (Figure 6).

### 3.4. Deriving Toxicity Measures from Bacterial Heat Flow Curves

The majority of IMC experiments address the inhibition of bacterial growth by evaluating the reduction of the initial exponential growth rate *r_i_* of a culture, given in dAR-TS by:(14)$$r_{i}=\frac{dP}{dH}\left(0\right).$$

The most common descriptor of toxicity is the concentration of a compound which causes half-maximal inhibition (IC_50_). However, reporting an IC_50_ for IMC data is ambiguous with respect to the choice of the observables *r_i_*, *P_m_*, *H_m_* or *t_m_*, which may be redundant or report different aspects of toxicity. With the widespread and intuitive use of the initial growth rate as a descriptor of toxicity, it is also reasonable to use *r_i_* for the evaluation of toxicities in dAR-TS but complement it with Θ to express the complexity of toxic effects revealed in IMC measurements on metal toxicity. Figure 7 shows a “rate plot”, which displays the initial growth rate *r_i_* as a function of the initial metabolic load Θ(0). It is convenient to choose a half-maximal initial growth, i.e., a metabolic load of Θ(0) *=* 0.5, as the standard growth condition to compare thermograms from a culture with a “stressor” to a reference IMC trace without a “stressor”. Correspondingly, the “standard initial growth rate” is *r_i_ =* 0.5*∙r*_0_. Defining the maximal growth rate *r*_0_ of a control experiment as unity, the “standard initial growth rate” equals 0.5. This defines the reference point [0.5;0.5] in the “rate plot” (Figure 7, black-filled circle). Transforming IMC data to the standard conditions (see Methods) renders the “rate plot” invariant against medium concentration and ampoule volume, such that it displays metabolic differences exclusively.

Figure 7 visualizes the different effects on the growth of *L. lactis* exerted by copper, uranyl and temperature shown in the previous sections. Toxicity scales with the displacement of the [*r_i_*;Θ(0)] data pairs from the reference point [0.5;0.5]. The plot of the uranyl data represents three fundamental forms of toxicity: (a) predominant attenuation of the maximal growth rate causes the mostly vertical (negative) displacement of the data pairs (third growth phase, open red triangles; (b) predominant attenuation of the metabolic load causes a diagonal displacement (because *r_i_* scales with Θ(0), second growth phase, open red circles); (c) compensation of increased maximal growth rate and decreased “metabolic load” causes an adaptive response (initial growth rate almost unaffected, first growth phase, filled red circles). For copper, the plot indicates that the bacteria made up for the reduction of the maximal growth rate (vertical displacement) by an increase of their apparent nutrient affinity (increased “metabolic load”). The metal-induced displacements of the data points are systematic for each growth phase, supporting again the physiological relevance of the dAR-TS parameters. The validity of this concept is convincingly confirmed by the temperature experiment (Figure 4c,d). All metabolic phases experienced the expected increase in maximal growth rates with negligible effect on the “metabolic load”, i.e., an empirical correlate of the constancy of growth–metabolism relations. With respect to the transferability of toxicity measures between labs, the “rate plot” can provide a convenient IC_50_, for example, by the manifold of growth-inhibiting conditions given by all data pairs that are displaced from the reference point by a distance of log2.

## 4. Discussion

The simulation of metabolic heat flow data by dAR-TS assigns “canonical thermograms” to distinct metabolic phases of cultured cells. It describes heat release during growth and metabolic decline for dividing and non-dividing cells as well as for variable inoculate sizes according to Equation (1a). The “metabolic load” (Equation (1b), Figure 3a) expresses the nutrient dependence of metabolism by the concentration *x* of nutrients in proportion to x/(x+c), as does the (empirical) Monod equation. In contrast to the latter, it also applies to unbalanced growth. Remarkably, the investigated systems showed conserved substructures in their time-dependent thermograms. These structures become analytically accessible in the above hyperbolic form when expressed as a function of the total heat release. Defining a metabolic phase as the constancy of a predominant biochemical pathway of nutrient consumption during culture growth, then the total heat release is a measure of total biomass and thus of the (increasing) catalytic capacity of the culture. The hyperbolic substrate-dependence of exponential growth, which was formulated originally for exponential bacterial growth by Monod [1], can, in fact, be extended far beyond the balanced regime when the increase in catalytic capacity during growth is taken into account.

One reason for the reproduction of the rather complex metabolic activities of *L. lactis* by dAR-TS lies in the metabolic switching processes which occur long before the quasi-steady state condition of “nutrient adsorption” in the dAR-TS formalism would be violated by substrate depletion. The observed physiologic switches between different enzymatic machineries, (evidenced by the change in the metal sensitivity of the growth), occur shortly after the maximal metabolic activity has been reached for a given metabolic phase. In all traces, the physiological reorientation occurred at metabolic loads larger than 0.5, i.e., the active pathway always stayed above half saturation with nutrients. Only in the late metabolic phase(s) where stress-related metabolic pathways become activated, the assumption of an apparent equilibrium of nutrient adsorption followed by catalytic turnover does not hold for the *L. lactis* culture. However, the *T. congolese* data show convincingly that the hyperbolic nutrient dependence assumed in dAR-TS does correctly describe cell growth from exponential growth to complete nutrient consumption in the absence of such physiologic adaptations upon nutrient depletion.

Analogous to extending the hyperbolic Monod equation to unbalanced growth by accounting for biomass increase and nutrient depletion, a similar mathematical treatment of the hyperbolic Michaelis–Menten (MM) equation should hold equally well. In this case, the constant enzyme concentration of the MM model must be allowed to grow in proportion to catalytic turnover to describe culture growth. It is thus instructive to consider the closed form of the integrated Equation (13), i.e., the “canonical thermogram” in the time domain, which is obtained as:(15)$$H\left(t\right)=H_{0}-H_{f}\cdot \;W\left(\frac{H_{0}}{H_{f}}\cdot e^{\frac{H_{0}-r{\;}_{0}\cdot H_{\emptyset }\left(t\right)\cdot t}{H_{f}}}\right)$$
with $H_{\emptyset }\left(t\right)$ being the time-averaged metabolic heat and W being the Lambert Ω-function (Methods). It should not be surprising that the closed form of the classical integrated (quasi-static) MM equation exhibits the same structure [39]. The quantities *H*_0_, *H_f_* and *r*_0_ are mathematical homomorphs of the initial substrate concentration *S*_0_, MM constant *K_M_* and enzyme catalytic rate *k_cat_*, respectively. The term $r_{0}\cdot H_{\emptyset }\left(t\right)$ replaces the expression $k_{cat}\cdot E_{0}$ for the maximal catalytic rate *V_max_* at enzyme concentration *E*_0_. This reveals the formal correlation of the constant *E*_0_ in MM kinetics with the time-averaged biomass, here in the form of its heat equivalent $H_{\emptyset }\left(t\right)$. It is evident that dAR-TS is formally a generalization of the MM equation, but it does not imply mechanistic assumptions other than the proportionality between cell number and catalytic capacity of cultured cells, where the metabolic activity of each cell follows a hyperbolic nutrient dependence. This agrees with non-calorimetric analyses of limited bacterial growth, where growth rates have been plotted as a function of biomass, rather than culture time, analogous to the “enthalpy plot” introduced here [40].

The dissection of complex thermograms into simpler “canonical thermograms” by dAR-TS may appear artificial because it supposes a heterogeneity of metabolic states without further justification. It is indeed the experimental evidence provided by the demonstrated phase-specific metal toxicities, which lends physiological relevance to this dissection and, at the same time, strongly supports the prevalence of simple growth–metabolism relations in these phases. Thus, the dynamic Langmuir adsorption reaction model of the growth–metabolism relationship is not only physiologically appropriate, but also analytically tractable, because it reduces a complex growth behavior into sections of “two-parameter-fits”, rather than resorting to a many parameter problem in a single equation with little relation to physiology. Furthermore, the proposed standardized evaluation and graphical representation of metal toxicity developed here for IMC data is applicable for any other experimental method as long as an absolute growth rate (metabolic thermal power) can be determined as a function of culture biomass (heat) at a sufficient signal to noise ratio. Likewise, the two-dimensional representation of toxicity in a “rate plot” is not restricted to IMC data and represents the minimum number of parameters (*r_i_*, Θ) that express the experimentally evident diversity of growth-inhibitory effects.

The salient simplification of growth analysis by dAR-TS stems from the transformation of thermograms (or absolute growth rates) from time to enthalpy (or biomass) domain. Thereby, the complexity of the non-analytical expression of *H*(*t*) is removed, showing that growth models can imply simple growth–metabolism relations despite a complex time-dependence. This raises the question of to which extent common growth models can cover the conserved “dynamic Langmuir-like” growth–metabolism relations demonstrated here. Among rate equations derived for prominent growth models [41], the Richards function can be parametrized to resemble a Monod or MM equation (in line with the statistical superiority of the Richards function over Gompertz and logistic growth models) [6]. Expressing the Richards growth curve in heat equivalents is compatible with dAR-TS and approximates the growth of *T. congolese* well by adjusting the Richards function exponent *m* (Figure 8). For the *T. congolese* culture, a “shape parameter” of 0.67 is obtained from dAR-TS, whereas it is predicted to be 0.61 by the Richards Function. The deviation is due to an overestimation of the maximally released heat by the Richards function as compared to a ”dynamic Langmuir-like” growth curve (Figure 8). Nevertheless, dAR-TS parameters for the description of the growth of cultured cells can be obtained from the Richards function by substituting *s_pR_* for *s_p_* (Equations (4a)–(4d)).

Interestingly, the Baranyi growth model uses an “adjustment function” α(*t*) similar to the “metabolic load” Θ(*H*) [42]. However, the MM homomorphism is disfavored for the sake of simplicity and strength in the lag time simulation of bacterial growth [43].

In the context of existing analytical growth models, the most biologically relevant prediction of dAR-TS concerns the maximal absolute growth rate, which cannot be reached before 50% of the final biomass of cultured cells have been produced. In enthalpies: the heat *H_m_* released at maximal metabolic activity is at least 50% of the heat *H*_0_ released upon full nutrient consumption in the considered metabolic state. This follows from the relation:(16)$$1>s_{p}=\frac{H_{m}}{H_{0}}>0.5$$
as derived above (Equation (3c)). In contrast, Gompertz and logistic growth curves exhibit their inflection points invariably at 37% and 50%, respectively [44]. Importantly, the thermogram “shape parameter” for the initial culture growth phase is fully determined by the experiment as a descriptive parameter without any mechanistic assumptions or restrictions of its absolute value, which is key to thermogram interpretation and data transferability between labs (Equation (7)). Only in the context of dAR-TS, the absolute value of *s_p_* becomes restricted by the underlying Langmuir adsorption catalytic turnover formalism. More remarkable is the observation that the discussed thermograms and all others recorded thus far in this lab complied with the prediction from Equation (16). Furthermore, the dynamic adsorption reaction model is compatible with advanced thermodynamic [13], kinetic (including flux balance analyses) [7,12,45,46] and proteome allocation-based growth laws [8,47,48], which lead to hyperbolic nutrient dependencies of (balanced) growth as well. This enables analytical approaches to advanced growth–metabolism relations throughout the entire lifespan of cultured microorganisms without the approximations required for an analytical solution of the Monod Equation in the time domain [49]. Promising stochastic descriptions of the fundamental biochemical processes underlying metabolism and growth are currently emerging but, again, suffer from the restriction to balanced growth [50].

## 5. Conclusions

Metabolism and growth of cultured cells from diverse origins follow a succession of relatively simple activity patterns that are not reproduced by prominent analytical time-dependent growth models. On the other hand, it has been shown here that the hyperbolic nutrient dependence of growth originally formulated by Monod can be easily extended to the unbalanced regime when growth is expressed as a function of biomass (here expressed as enthalpy), rather than time. The mathematical equivalence with an extended MM formalism may or may not imply metabolic phases, where unbalanced growth is also restricted by a single rate-limiting “master enzyme. The success of the proposed “dynamic Langmuir adsorption reaction” approach rather raises the question of whether more complex stochastic growth models will eventually explain why the kinetics of a complex metabolic network may nevertheless follow a simple hyperbolic nutrient dependence. The unprecedented accuracy and high information content of IMC data will probably allow for the testing of advanced stochastic theories of growth–metabolism relations by experimentation.

## Figures and Tables

**Figure 1 microorganisms-10-01397-f001:**
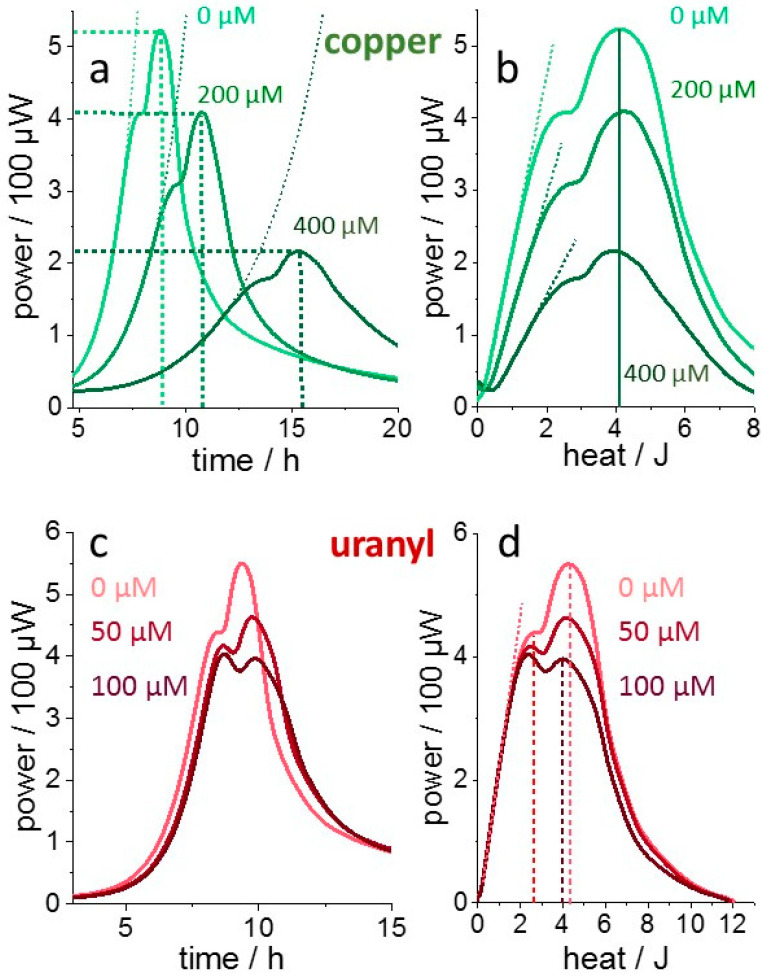
Transformation of thermograms from the time to enthalpy domain. (**a**), Time-dependent thermograms of a glutathione-expressing *L. lactis* culture grown in the presence of the indicated copper sulfate concentrations. The exponential growth is highlighted by exponential functions (dotted lines). (**b**), “Enthalpy plot“, the same traces plotted dependent on the totally released heat, where exponential growth corresponds to a linear dependence (dotted lines) of thermal metabolic power on total released heat which scales linearly with total biomass. (**c**,**d**), the equivalent data recorded from the same strain in the presence of uranyl nitrate [36].

**Figure 2 microorganisms-10-01397-f002:**
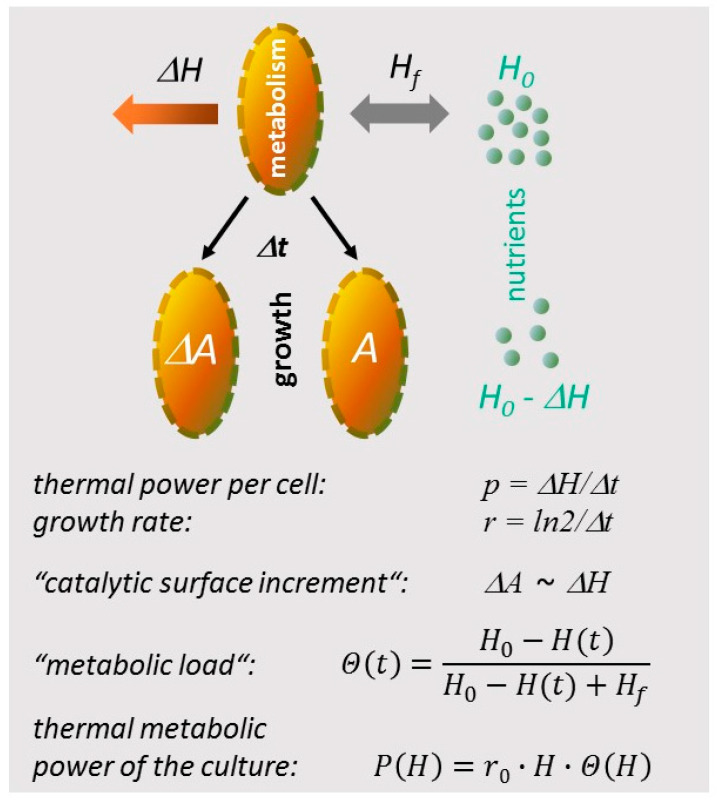
Representation of metabolism and growth in metabolic heat equivalents. *H*_0_: heat equivalent produced by the initial amount of nutrients if metabolized in a single biochemical pathway by cultured cells. *H_f_*: heat equivalent of the amount of nutrients that generates half-maximal metabolic thermal power (analogous to a Langmuir adsorption constant describing half-maximal binding site occupation on a catalytic surface A). Δ*H*: heat released by a cell during the time Δ*t* between cell divisions (symbolized by arrows). Δ*A*: increment in apparently catalytic surface A due to formation of a new cell. *H*(*t*): accumulated released heat up to time *t* in proportion to the amount of biomass; the latter provides an apparent nutrient-adsorbing total catalytic surface *A*. Θ: “metabolic load” describing the fraction of occupied nutrient-binding sites on *A* (*H*_0_
*− H*(*t*) scales with the amount of nutrients at time *t*). *P*: thermal power of a culture of microorganisms expressed as the product of released heat H, maximal growth rate *r*_0_ (Methods, Equations (1a) and (1b) and condition for Equation (3c)) and ”metabolic load” Θ.

**Figure 3 microorganisms-10-01397-f003:**
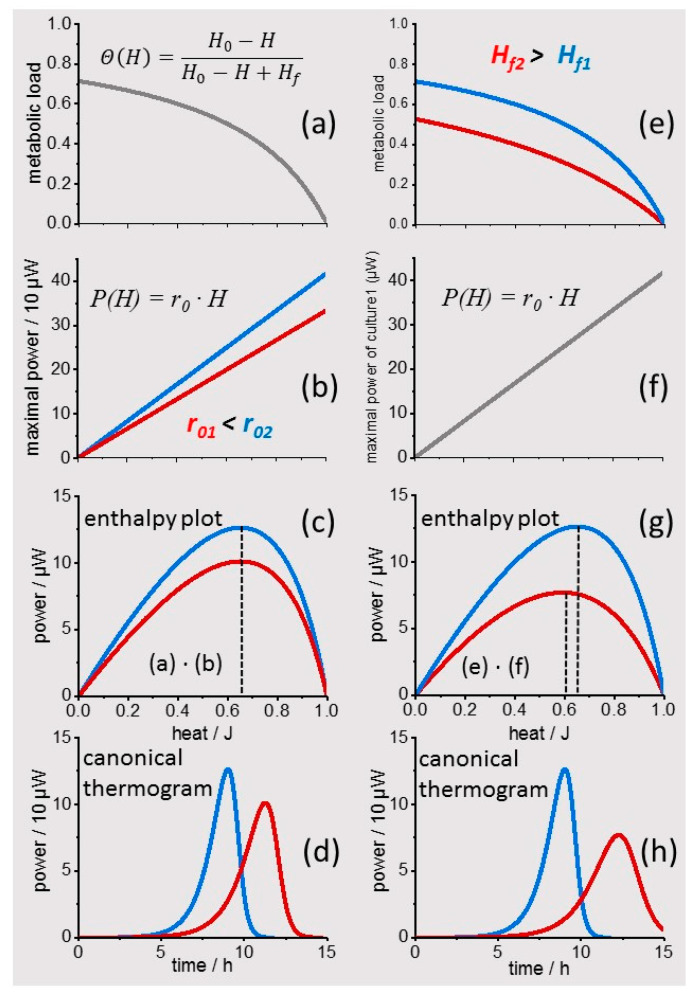
Generation of “canonical thermograms”. (**a**), “Metabolic load“ Θ expressed in the form of a Langmuir adsorption isotherm as a function of the total enthaply *H* released from cultured cells. *H* is obtained in the experiment from the integration of the thermal power over time. (**b**), Linear relation between the apparent nutrient-adsorbing catalytic surface (in proportion to *H*) and the metabolic thermal power *P* of the culture (two examples shown in red and blue). (**c**), “enthalpy plot“ generated by multiplication of the traces in (**a**,**b**). (**d**), “canonical thermograms“ obtained by numerical integration of the inverse of the traces in (**c**). (**e**–**h**), same operations as in (**a**–**d**) with two Langmuir adsorption isotherms differing in *H_f_*, which leads to a shift in the peak of the “enthalpy plot” (dotted lines).

**Figure 4 microorganisms-10-01397-f004:**
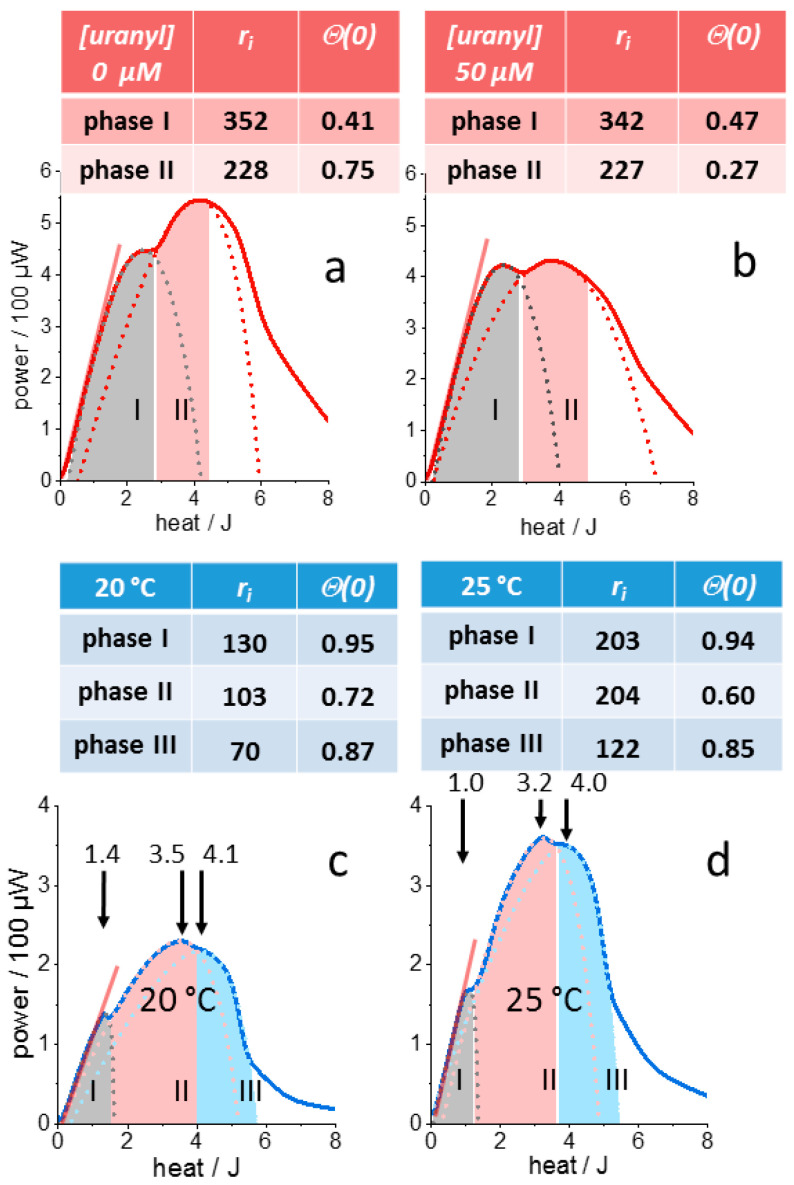
dAR-TS analysis of growth-attenuating and -stimulating effects on *L. lactis* cultures. (**a**), Initial growth rate *r_i_* and initial metabolic load Θ(0) for the first two consecutive metabolic phases (grey and pink shaded areas). (**b**), Culture grown from the same inoculate but in the presence of 50 µM uranyl. The latter affects the second metabolic phase by an increase of *H_f_* (thus reducing the initial “metabolic load” Θ(0)) at a constant initial growth rate. The obtained initial rate *r_i_* and “metabolic load” Θ(0) vary by less than ±5% for the varying of *P_m_* by ±1 SD. For only the fit of the second metabolic phase with 50 µM uranyl, Θ(0) varies by 20%. (**c**), *L. lactis* culture grown at 20 °C, showing the capability of dAR-TS to reproduce and also the decline of metabolic activity in the third metabolic phase (light blue dotted line, blue-shaded area). (**d**), Thermogram of the same strain grown at 25 °C. Straight lines (red) mark the exponential growth phase of the initial metabolic state. Units of *r_i_* is 10^−3^ h^−1^. The standard deviation of the fitted curves (dotted lines) from the raw data is smaller than 1.5% of *P_m_* for all traces. The obtained initial rate *r_i_* and “metabolic load” Θ(0) vary by less than ±3.5% for a variation of *P_m_* by ±1 SD. Only for the fit of the second metabolic phase at 25 °C (pink dotted line in (**d**)), *r_i_* and Θ(0) vary by ±9% and ±20%, respectively.

**Figure 5 microorganisms-10-01397-f005:**
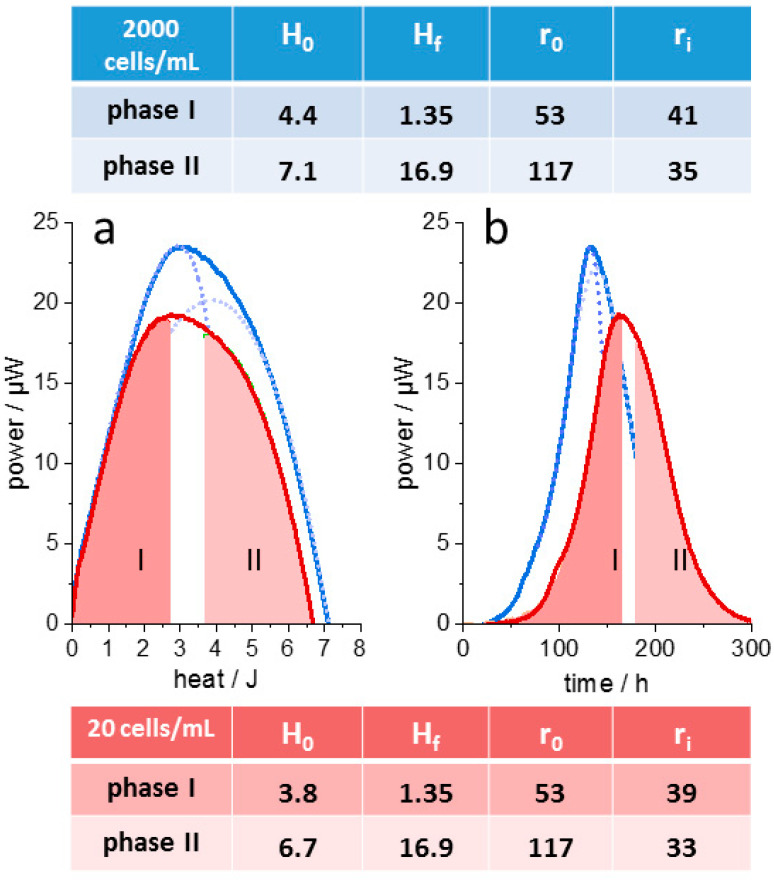
Thermograms of Trypanosoma congolese. (**a**), Inoculate of 20 cells (**red**) and 2000 cells (**blue**) per mL. For the culture grown from the smaller inoculate, the “canonical thermograms“ are displayed as colored areas corresponding to the first (ocre) and second metabolic state (**green**). Dotted lines show the two “canonical thermograms“ obtained with the larger inoculate. (**b**), Numerical transformation of the “canonical thermograms” from (**a**) the time domain (dotted and dashed curves color-coded as in (**a**)) superimposed with the original IMC raw data (solid lines). Original traces have been published [38]. The listed thermogram parameters were kept constant, except for H0 and N0, which were changed between the two data sets and substituted in Equation (1a) with *p*_0_ = 3.2 µJ/h and *E* = 16.667/J. Growth rates *r*_0_ and *r_i_* in 10^−3^h^−1^, *H*_0_ and *H_f_* in Joules. Standard deviation of fitted data range (overlap region of blue and red trace) from raw data is less than 1% of peak power.

**Figure 6 microorganisms-10-01397-f006:**
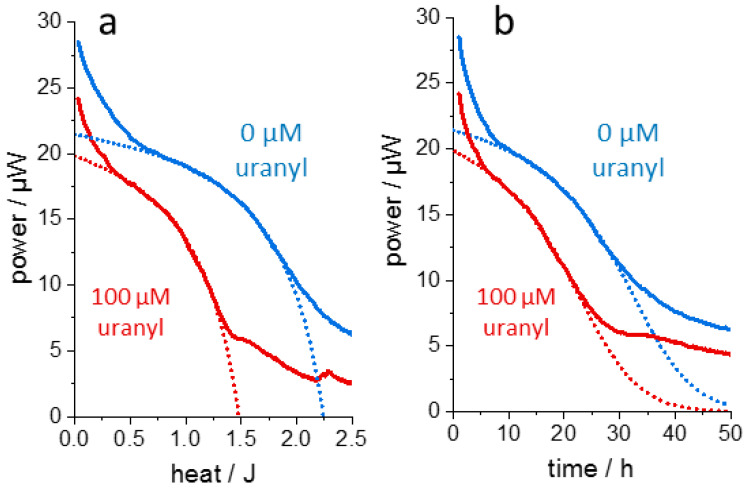
Thermograms from plant callus cells (*Brassica napus*). The non-dividing plant cells were cultured in the absence and presence of 100 µM uranyl nitrate and the enthalpy plots ((**a**)**,** solid lines) generated from the raw data ((**b**), solid lines). Thermogram analysis by dAR-TS reveals that after an adaptation phase of about 8 h, for about 12 h the cultures followed the assumed Langmuir–isotherm relation in the enthalpy domain and the corresponding Lambert Ω functions in the time domain (dotted lines) described by the “metabolic load” Θ for non-growing cells (*E =* 0, see Equations (1a) and (1b) in the main text). The effect of uranyl is well reproduced mainly by reducing the maximally released heat *H*_0_ from 2.25 J in the reference thermogram to 1.49 J for the thermogram obtained in the presence of uranyl (*x*-axis intercepts). A possible explanation is the formation of insoluble or non-metabolizable sugar uranyl complexes which diminish the total amount of available nutrient. The raw data have been published previously [37].

**Figure 7 microorganisms-10-01397-f007:**
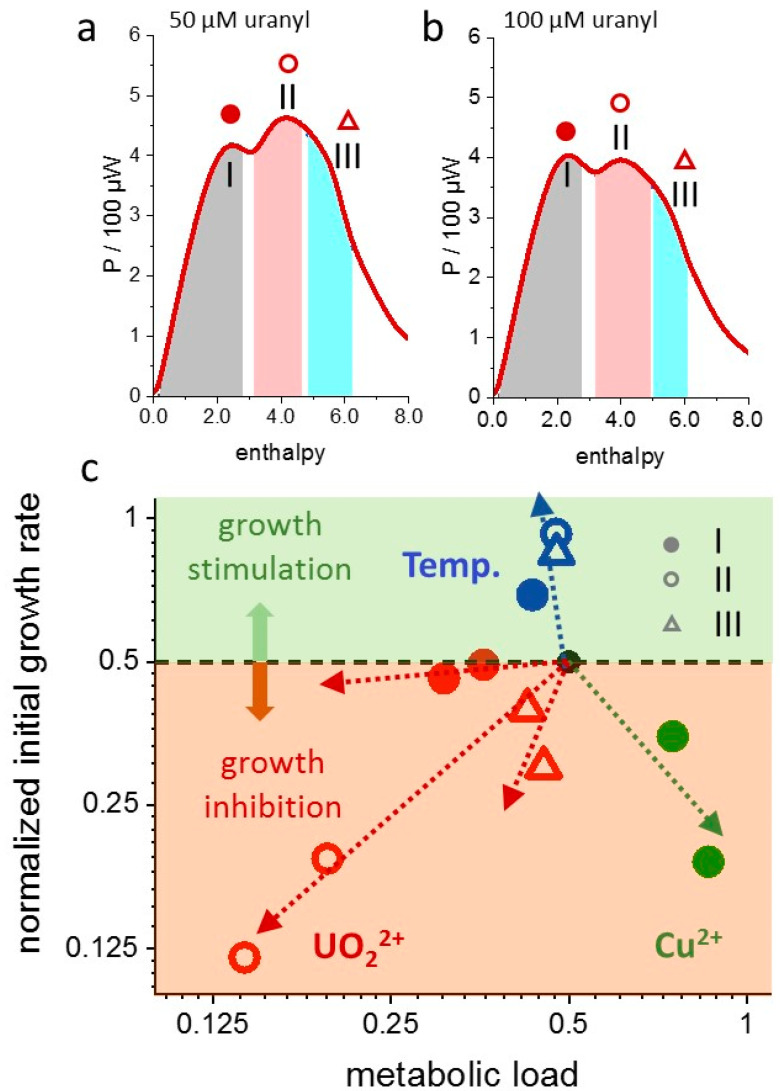
Generation of a “rate plot” for *L. lactis* grown in the presence of uranyl or copper. (**a**,**b**), “Enthalpy plots“ of a glutathione-synthesizing *L. lactis* strain grown in the presence of 50 µM or 100 µM uranyl, respectively. Shapes of shaded areas calculated by dAR-TS. (**c**), “rate plot“ showing the metal-dependent change in initial rate *r_i_* = *r*_0_∙Θ(0) and “metabolic load” Θ(0), normalized to the condition of Θ(0) = 0.5 and *r*_0_ = 1 without metals (black-filled circle). Red symbols: “canonical thermogram“ parameters used in (**a**,**b**) with symbols assigned to growth phases I, II, III as depicted, green symbols: “canonical thermogram“ parameters of the first growth phase in the presence of copper (thermograms in Figure 1a). Arrows indicate the direction of increasing metal concentration (50 µM and 100 µM for uranyl and 200 µM and 400 µM for copper). Colored regions correspond to an increased (light green) or decreased (light orange) initial growth rate relative to that of a reference culture.

**Figure 8 microorganisms-10-01397-f008:**
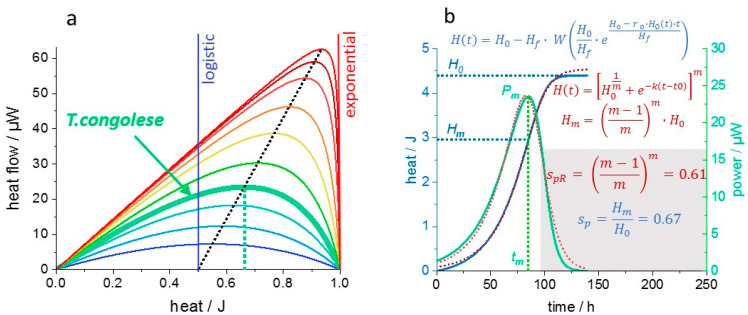
Comparison of thermograms simulated by dAR-TS and by a Richards growth curve. (**a**), “Enthalpy plots” of thermograms according to Equations (3) and (4) with (*H*_0_
*=* 1 and *H_f_ =* 2, 1, 0.55, 0.35, 0.2, 0.1, 0.05, 0.02, 0.01, 0.005 from blue to red). The thermogram peak height increases linearly with the “shape parameter”. At a very low nutrient affinity, the thermogram shape is a symmetric parabola and corresponds to logistic growth in the time domain. At high nutrient affinity (*H_f_ « H*_0_), *s_p_* approaches unity and the growth is purely exponential. The “enthalpy plot” of the first growth phase of *T. congolese* (Figure 5a) is indicated. (**b**), The *T. congolese* time domain “canonical thermogram” of the first growth phase (2000 cells/mL, *s_p_ =* 0.67, turquoise) has been integrated to obtain a corresponding growth curve (blue solid sigmoid ). To the latter, a Richards function (red dotted line) was fitted (Richards1 in Origin (Origin(Pro), 2020. OriginLab Corporation, Northampton, MA, USA) with *H*_0_
*=* 4.54 and *m = −*0.3957. The corresponding “Richards thermogram” (red dotted bell-shaped curve) was obtained as the derivative of the Richards function. The respective formulas are given in color code (dAR-TS: blue, Richards: red) with *s_pR_*: the shape parameter obtained from the Richards curve.

## Data Availability

Data in Figure 4c,d are available on request from the author.

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
