# Peer review of "Simple Growth–Metabolism Relations Are Revealed by Conserved Patterns of Heat Flow from Cultured Microorganisms"

_microorganisms, 2022, doi:10.3390/microorganisms10071397_

Round 1

Reviewer 1 Report

In this contribution, the author offers a radically new and surprisingly simple quantitative relationship between microbial growth and metabolism as gauged by calorimetric measurements of heat release. While such measurements are not new (in fact, they marked the early days of biocalorimetry during the times of Lavoisier, when animals were used instead of microbes), the author here suggests a fundamentally different and refreshing way of analyzing and interpreting such measurements quantitatively. This is accomplished with the aid of so-called “canonical thermograms”, which provide a unifying picture (see, e.g., Fig. 1b and 1d) of metabolic processes that, at first glance, might appear intimidatingly complex. Fig. 3 summarizes the workflow for the generation of such “canonical thermograms” in a pictorial manner, which is very helpful, particularly for those readers not willing to go through the mathematical details required for quantitative analysis. The author’s analyses and interpretations are thorough and sound, and the manuscript is written very clearly.

My only (minor) concern relates to the use of the term “Michaelis-Menten-like”, particularly in connection with mechanistic explanations. The saturation behavior characteristic of MM enzyme kinetics is not a peculiarity of this model but is found across a much broader class of models that describe or depend on saturable binding. In fact, MM inherits its hyperbolic “saturation curve” shape from the postulate that an enzyme/substrate complex needs to be formed before enzymatic catalysis can take place. Moreover, it is known that many—if not most—enzymes exhibit more complex mechanisms than simple MM but, at least superficially, can still approximately be described using a “Michaelis-Menten-like” mathematical form. Therefore, a broader term such as “Langmuir adsorption isotherm” (as also used a number of times by the author) or simply “saturation behavior” might be more appropriate.

Minor comments:

Line 84: “exception” rather than “exemption”?

Line 139: “Simulation startS…” or “SimulationS start…”.

Caption to Fig. 1: The colors mentioned in the figure caption (black, red, blue) do not match the colors used in the figure panel. In general, it would be easier for the reader if the curves were labeled directly within the figure panel without the need to read the caption to decipher the color code.

Figure 8b appears rather crowded, and I am not sure if the equations can be read unless this figure and the fonts used in it are enlarged by at least 20%.

Author Response

response to Reviewer 1

I thank the reviewer for the appreciation of the work and the positive comments on the novelty of the approach.

  1. In accordance with the suggestion to avoid the term “Michaelis-Menten-like” in the context of growth and metabolism, I have removed the term throughout the manuscript and replaced it by either “dynamic Langmuir adsorption” or “hyperbolic nutrient dependence”, including the abstract (line 25 clean manuscript pdf version).

Reference to the Michaelis-Menten (MM) equation is restricted to the comparison of the mathematical structure of the closed form of the integrated MM-equation with that of the corresponding closed form of the “dynamic Langmuir adsorption” formalism. The relation to the Michaelis-Menten equation is now restricted to te discussion only (line 546-568) and mentioned accordingly in the abstract (line 27-29, clean manuscript pdf version).

At various places in the original manuscript it had been made clear already that mechanistic assumptions on growth-metabolism relations are assumed neither in dAR-TS nor by the formal similarity with the hyperbolic substrate dependence of enzyme activity expressed by the MM-equation. Nevertheless, the reader may grow more familiar with the two parameter description (“metabolic load” and “apparent nutrient affinity”) of culture growth, when the structural similarity between the two formalisms is detailed and the consequence of their conceptual difference documented.

Likewise, the multitude of mechanistic models of cell growth that lead to hyperbolic nutrient dependencies, as mentioned by the referee, is now more clearly expressed by moving the corresponding comments and references from the conclusions to the final paragraph of the discussion (line 635-643, clean manuscript pdf version).

  1. typos have been corrected
  2. Labels for concentrations have been introduced in Figure 1 directly as proposed, color code has been removed from legend
  3. font size of equations in Figure 8b has been increased

With the proposed changes, the manuscript has gained clarity in its focus which lies on providing an evaluation tool for repetitive thermogram structures and there use in toxicity assessment, rather than suggesting mechanistic interpretations. Therefore, I hope you will find the manuscript acceptable for publication.

With kind regards

Fahmy

Reviewer 2 Report

Dear Authors

Please revise your manuscript according to my comments in the attached pdf file

Author Response

response to Reviewer 2

I thank the reviewer for the detailed comments and suggestions which have all been taken into account in the revised version of the manuscript as described below.

1. the title has been changed to: “Simple growth-metabolism relations are revealed by conserved patterns of heat flow from cultured microorganisms”

- in contrast to the proposed term “heat emission”, “heat flow” is a well-established expression in the bio-calorimetry field

2. First sentence of introduction has been rephrased as proposed. (line 37 in the clean pdf version of the manuscript)

3. Sentence has been rephrased as proposed (line 60-63 in the clean pdf version of the manuscript)

4. subtitle 2.1 has been shortened to: 2.1. Heat flow curves from L.lactis at different temperatures

5.-7. formatting has been corrected (line 108 in the clean pdf version of the manuscript)

8. caption Fig. 6 has been lowered

9. caption Fig. 7 has been shortened

10. Eq. 17 has been centered

11. References have been moved from the conclusion section to the end of the discussion as proposed, the conclusion has been slightly rephrased accordingly.

- The design of the evaluated experiments is not subject to revisions as the majority of the data has been taken from previously published work and the novel experiments on temperature-sensitive growth of L.lactis follows these same published procedures for comparability.

- To the best of my knowledge, the introduction covers the most relevant literature for the scope of the manuscript, i.e. proposing an evaluation tool for IMC data from cultured microorganisms to address growth metabolisms relations and derive toxicity measures. Rather than testing hypotheses on specific biochemical pathways, divers systems have been chosen to assess the generalizability of the approach.   

The changes proposed by the referee have certainly improved the readability of the manuscript which I hope you will find suitable for publication.

With kind regards

Karim Fahmy